

# The lapse rate cold point tropopause in the Asian Summer Monsoon anticyclone

Rolf Müller[1], Bärbel Vogel[1], Martina Krämer[1], Christian Rolf[1], Nicole Spelten[1], and Fabrizio Ravegnani[2]

[1]Institute of Climate and Energy Systems (ICE-4), Forschungszentrum Jülich, 52425 Jülich, Germany
[2]National Research Council – Institute for Atmospheric Sciences and Climate (ISAC-CNR), 40129 Bologna, Italy

**Correspondence:** Rolf Müller (ro.mueller@fz-juelich.de)

**Abstract.** Tropospheric and stratospheric airmasses are separated by the tropopause. Here we investigate the lapse rate tropopause and the cold point tropopause in the Asian summer monsoon anticyclone (ASMA) based on high-altitude airborne measurements in summer 2017. We find that, in the ASMA, the lapse rate tropopause, and not the cold point, constitutes a good estimate of the upper boundary of the well mixed tropospheric air mass. On average the cold point is located about 1 km above the lapse rate tropopause and is about about 3 K colder with the pressure lower by about 12 hPa. Above the cold point tropopause molar water vapour mixing ratios range between ∼3 and 10 ppm. Molar ozone mixing ratios increase substantially with altitude; between the lapse rate tropopause and the cold point tropopause molar ozone mixing ratios are in the range of 50-200 ppb. There is slow, diabatic, upward transport in the vicinity of the lapse rate tropopause and above. Hydration patches above the cold point tropopause were observed with water vapour mixing ratios of ≲ 10 ppm. No indication of substantial dehydration above the cold point tropopause in the ASMA was found in the observations. For strong convection (flight on 10 August 2017) there is substantial dehydration at the cold point tropopause (indicated by high values of total water, ice particle occurrence, and strong supersaturation). Above the cold point tropopause, under such conditions, neither ice particle occurrence, nor enhanced molar mixing ratios of water vapour (above ≲ 6 ppm) are observed.

## 1 Introduction

In the tropics, atmospheric composition in general and the temperature profile in particular is rather different above and below the tropopause (e.g., Hoskins and Rodwell, 1995; Park et al., 2007; Bian et al., 2012; Pan et al., 2014, 2018; Krämer et al., 2020). Below the tropical tropopause the atmosphere is of tropospheric nature and is vertically strongly mixed, above the tropopause the air is more of stratospheric nature; in particular the air shows lower mixing ratios of water vapour and higher mixing ratios of ozone (e.g., Jeffery et al., 2022).

The tropopause is classically defined by the atmospheric lapse rate (WMO, 1957) (i.e., the location of a significant change in the atmospheric temperature lapse rate); the lapse rate tropopause is also referred to as the thermal tropopause. Overshoots of convection across the lapse rate tropopause occur (e.g., Khaykin et al., 2022; Clapp et al., 2023), in particular in the monsoon region. This leads to transport of water vapour across the tropopause into the lowermost stratosphere. The transport of



water vapour into the lowermost stratosphere and the abundance of water vapour in the lower stratosphere are important for tropospheric climate and dynamics; processes in this regard are often not well simulated in models (Solomon et al., 2010; Charlesworth et al., 2023; Ploeger et al., 2024).

The tropopause is a concept that has long been established (e.g., Hoinka, 1997), but which is still a topic of current research (e.g., Kunz et al., 2011; Pan et al., 2014, 2018; Maddox and Mullendore, 2018; Hoffmann and Spang, 2022; Tinney et al., 2022; Zou et al., 2023; Zhang et al., 2024; Turhal et al., 2024; Reutter and Spichtinger, 2025; Bauchinger et al., 2025; Riese et al.,

2025). There are different ways to determine the location of the tropopause (e.g., Hoinka, 1997; Kunz et al., 2011; Maddox and Mullendore, 2018; Pan et al., 2018; Turhal et al., 2024; Reutter and Spichtinger, 2025; Bauchinger et al., 2025). There are also variations over long time scales (decades) of tropopause height and temperature (e.g., Zou et al., 2023).

Often, the location of the tropopause is considered as a cold point (the point in the profile showing the lowest temperature) or as the lapse rate tropopause (WMO, 1957). These two different ways of locating the tropopause lead to different conclusions

on the properties of air masses in the vicinity of the tropopause (Munchak and Pan, 2014; Pan et al., 2018). The cold point and the lapse rate tropopause are sometimes collocated, but often the tropical cold point is located substantially above the lapse rate tropopause (Munchak and Pan, 2014; Pan et al., 2018). Both the cold point and the lapse rate tropopause were analysed based on aircraft measurements in January-March 2014 in the western Pacific (Pan et al., 2018); finding that the air is drier on average at the cold point than at the lapse rate tropopause. Further, Pan et al. (2018) found a significant air mass discontinuity at the

lapse rate tropopause. They conclude that there is no local barrier for transport across at the lapse rate tropopause, but rather that this boundary reflects the with altitude diminishing role of convectively driven transport and mixing. This is important as the tropical west Pacific in winter is a key entry region for tropospheric air into the stratosphere. (e.g., Sun et al., 2025).

Likewise, the Asian monsoon anticyclone (ASMA) is an important entry region to the stratosphere which will be investigated here. We use measurements on board of a high flying research aircraft in July and August 2017 (that is during the Asian summer

monsoon peak season), during the StratoClim campaign (e.g., Legras and Bucci, 2020; Krämer et al., 2020; Singer et al., 2022). Trace gas, water vapour, and particle measurements during this campaign have been analysed before (e.g., Höpfner et al., 2019; Krämer et al., 2020; Legras and Bucci, 2020; von Hobe et al., 2021; Singer et al., 2022; Konopka et al., 2023; Vogel et al., 2023). Also, the moistening of the lower stratosphere in the ASMA observed in a specific flight (8 August 2017) during the StratoClim campaign was investigated (Lee et al., 2019; Khaykin et al., 2022). Here we focus on tropopause characteristics in

the ASMA region during the StratoClim campaign in summer 2017.

The ASMA exists in Northern hemisphere summer in the upper troposphere and lower stratosphere (e.g., Park et al., 2007; Gottschaldt et al., 2018; Fadnavis et al., 2023). The ASMA stretches over a large area, between the Eastern Mediterranean and Northern India and the Tibetan plateau (e.g., Park et al., 2007; Vogel et al., 2016; Manney et al., 2021; Becker et al., 2025; Kachula et al., 2025). In the Asian monsoon region and season, the lapse rate tropopause is typically located at greater altitudes

(i.e. at lower pressures) than the tropopause outside the monsoon region (by about 0.5 to 1 km or 20 K in potential temperature; Highwood and Hoskins, 1998; Bian et al., 2012; Ploeger et al., 2015; Vogel et al., 2015). Further, there is a strong year-to-year variability in altitude and the temperature of the lapse rate tropopause in the Asian monsoon region (Hoffmann and Spang, 2022; Zou et al., 2023).



Upward transport in the Asian summer monsoon is dominated by convective activity; there are convective sources of tro-
pospheric perturbations at lower altitudes (e.g., ozone, water vapour, CO, $CO_2$) which are detectable in the ASMA at greater
altitudes above about 370 K potential temperature (Park et al., 2007; Ploeger et al., 2013; Santee et al., 2017; Muhsin et al.,
2018; Ueyama et al., 2018; Legras and Bucci, 2020; Krämer et al., 2020; von Hobe et al., 2021; Singer et al., 2022; Vogel
et al., 2023, 2024; Clemens et al., 2024; Becker et al., 2025). When investigating the impact of convection on the stratospheric
composition in the ASMA it is important to consider the influence of transport across the lapse rate tropopause (Khaykin et al.,
2022; Clapp et al., 2023).

The impact of convective activity in the North American and Asian Monsoon regions on stratospheric humidity has been
investigated based on satellite observations (Randel et al., 2015; Wang et al., 2025); it is concluded that stronger convection
leads to a drier stratosphere, whereas weaker convection leads to a more humid stratosphere (Randel et al., 2015). Dehydration
in the upper troposphere and lower stratosphere over the Asian summer monsoon caused by tropical cyclones has also been
observed (Li et al., 2020). Further, moistening of the global lower stratosphere through sublimating ice particles (a process
showing a substantial regional variability) can be detected as an isotopic enhancement of the $HDO/H_2O$ ratio (Randel et al.,
2012; Singer et al., 2022).

Moreover, the altitude of the tropopause and its characteristics are very different in the extra-tropics and in the tropics; in the
tropics the transition from the troposphere to the stratosphere occurs in a layer, rather than at a sharply defined point referred to
as "tropopause". This fact has led to the concept of a tropical tropopause layer (TTL, Fueglistaler et al., 2009); with a bottom
at $\sim$ 150 hPa, 355 K, 14 km (pressure, potential temperature, and altitude) and a top at $\sim$ 70 hPa, 425 K, 18.5 km (Fueglistaler
et al., 2009). There is further a longitudinal variation of the tropopause altitude, in particular associated with the ASMA and
with El Niño-southern oscillation events (Gage and Reid, 1987; Highwood and Hoskins, 1998; Bian et al., 2012; Turhal et al.,
2024).

In the upper troposphere and lower stratosphere, temperature ($T$) altitude ($z$), pressure ($p$) and potential temperature ($\theta$) are
related (e.g., Krämer et al., 2020); here potential temperature $\theta$ is defined as

$$\theta = T \cdot \left( \frac{p_0}{p} \right)^{\kappa} \tag{1}$$

with $\kappa = R/c_p$, $R$ is the gas constant for dry air, $c_p$ is the specific heat capacity at constant pressure, and $p_0$ is a reference
pressure (1000 hPa for the calculations reported here). Pressure decreases with altitude and potential temperature increases
with altitude (see also appendix A). The relation in detail can be different above and below the tropical tropopause, where the
tropical tropopause temperature is close to the minimum in temperature (e.g., Pan et al., 2018; Hoffmann and Spang, 2022;
Singer et al., 2022). In any case, in discussions of Asian monsoon issues, altitude, pressure and potential temperature are all
used frequently; therefore we provide here relations between these quantities based on in-situ measurements in the ASMA in
2017 (see appendix A).

90 We focus here on airborne observations of water vapour, ozone, temperature, other meteorological parameters, and cloud oc-
currence (total water); all these quantities are accessible also through balloon-borne observations. Obviously, the measurements
during StratoClim provide a wealth of further information, which will be valuable for an extension of the present work.



Here we find that the lapse rate tropopause in the Asian summer monsoon anticyclone constitutes a good estimate for the upper boundary of the well mixed tropospheric air mass and that the lapse rate tropopause and the cold point tropopause are often not co-located; these two "tropopauses" are different. The cold point tropopause is in particular important for water vapour; in the ASMA, above the cold point tropopause, mixing ratios of water vapour are $\lesssim 10$ ppm. For strong convection there is substantial dehydration at the cold point tropopause and no indication of particle occurrence or enhanced mixing ratios of water vapour ($\gtrsim 6$ ppm) above the altitude of the cold point tropopause.

## 2 Measurements during the StratoClim campaign in the Asian Summer Monsoon from Kathmandu in 2017

### 2.1 Temperature, Altitude and Pressure

We use measurements of temperature on board the research aircraft by the scientific Rosemount probe (TDC, Shur et al., 2006). We use the Mach number corrected version of the TDC temperature measurements (Singer et al., 2022). Temperature accuracy and precision are 0.5 and 0.1 K, respectively (Singer et al., 2022). The altitude of the aircraft is deduced from the aircraft system (UCSE, Stefanutti et al., 1999); likewise pressure measurements are taken from UCSE, but there is no significant difference in the pressure measurements of the UCSE and TDC systems. The UCSE altitude is consistently used for all analyses throughout the paper.

### 2.2 Gas-phase water vapour, total water vapour and ozone

Gas-phase water vapour aboard the Geophysica was measured by the FLASH instrument (Sitnikov et al., 2007; Khaykin et al., 2022), a Lyman-$\alpha$ photofragment fluorescence instrument with 1 s time resolution, a precision of 0.2 ppm, and a measurement range of 1-1000 ppm. Total water vapour on board the aircraft is measured by the FISH instrument, which is also a Lyman-$\alpha$ photofragment fluorescence instrument using a forward facing inlet (Zöger et al., 1999; Meyer et al., 2015; Schiller, 2015). Overall, in the range from 4 to 1000 ppm, the total accuracy of FISH is 6–8 %; for lower mixing ratios down to 1 ppm, the uncertainty reaches a lower limit of 0.3 ppm (Meyer et al., 2015). Considering the enhanced sampling of cloud particles caused by the forward facing inlet of the FISH instrument, an estimate of the cloud ice water content in the atmosphere is possible (Schiller et al., 2008, 2009; Schiller, 2015; Afchine et al., 2018).

Gas-phase water vapour in clouds is taken from the FLASH measurements; because of the design of the inlet, FLASH only measures gas-phase water vapour in the atmosphere. If FLASH measurements are not available, gas-phase water vapour in ice clouds is approximated as saturation over ice. Outside of clouds FISH also reports gas-phase water vapour. A detailed comparison of gas-phase water vapour measurements of the FISH and FLASH instruments during the campaign in 2017 is reported elsewhere (Singer et al., 2022). Overall, gas-phase measurements (outside clouds) of the FISH instrument have shown good agreement with a number of independent aircraft hygrometers (Rollins et al., 2014; Singer et al., 2022).

We further use the relative humidity over ice based on the aircraft measurements. First, the water vapour saturation mixing ratio over ice is calculated from pressure and temperature aboard the aircraft. Second, the measured gas-phase mixing ratio





(here from the FLASH measurements) is used to calculate the ratio of gas-phase water vapour mixing ratio to the water vapour
125  saturation mixing ratio over ice, in other words the relative humidity over ice.

Ozone on board the aircraft is measured by FOZAN-II, a dual-channel automatic fast-response (1 s) chemiluminescent ozone
analyser (Ulanovsky et al., 2001). The measured ozone concentration range is 10-500 $\mu$g/m$^3$, the relative error is less than
10 % and the measurement range is from the ground to 22 km altitude (Ulanovsky et al., 2001). At 30 hPa (and 200 K) a range
of 10-500 $\mu$g/m$^3$ corresponds to a range in molar mixing ratio of ozone of 116 to 5776 ppb. The lower limit of the ozone
concentration measured by FOZAN-II (10 $\mu$g/m$^3$) means that the instrument can measure increasingly lower values of the
molar mixing ratio of ozone with increasing pressure; for example, at 1000 hPa 10 $\mu$g/m$^3$ corresponds to 3.5 ppb.

### 2.3  Water vapour isotopologues

In the StratoClim campaign, an indication of water vapour enhancements through sublimating ice particles is provided by
measurements of the HDO/H$_2$O ratio, which is provided by the Chicago Water Isotope Spectrometer (ChiWIS, Singer et al.,
2022). An isotopic enhancement of the HDO/H$_2$O ratio (i.e., heavy water vapour) indicates gas-phase water resulting from the
sublimation of ice particles (Hanisco et al., 2007). The ChiWIS measurements are not directly used here, but conclusions in
the literature on water vapour originating from sublimating ice particles. (Khaykin et al., 2022) are used. Values of the $\delta$D of
about $-450$ in ‰ are indicative of sublimating ice particles, where the ratio of HDO/H$_2$O is given in the $\delta$ notationy (Hanisco
et al., 2007): $\delta$D(‰) $= 1000 \times ((HDO/H_2O)/RVSMOW)$ (RVSMOW is the ratio of HDO/H$_2$O in Vienna standard mean
ocean water).

## 3  Results

### 3.1  Deducing the location of the tropopause

There is a variety of methods to determine the location of the tropopause (Hoinka, 1997); here we focus on two specific meth-
ods: the lapse rate tropopause and the cold point tropopause. Both methods require only a measurements of the atmospheric
temperature profile.

The lapse rate tropopause, is defined as the lowest level at which the temperature lapse rate decreases to 2 K/km or less,
provided that the average lapse rate between that level and all higher levels within 2 km does not exceed 2 K/km (WMO, 1957).
That is, a layer of 2 km above a candidate tropopause is required by this definition; so that the lapse rate tropopause is designed
for a vertical profile measurement. Thus, to determine the lapse rate tropopause, we select a part of the flight, here the first
three hours of the flight (i.e., the ascent of the aircraft, see Table 1 for details). For the same part of the flight, to facilitate a
comparison, we report values for the cold point tropopause (the point in the profile showing the lowest temperature); Table 1.
Values for the last part of the flight (descent into Kathmandu crossing the tropopause region) and the cold point tropopause
considering the *entire* flight are reported in the supplement.



**Table 1.** Overview of the StratoClim science flights 2017 from Kathmandu. Shown is information on the lapse rate (LR) and the cold point (CP) tropopause for selected parts of the flight (i.e. the first three hours of the flight; except F1, 1.9-3 hours of the flight and F7, 1-2.5 hours of the flight – in other words vertical profiles crossing the tropopause on ascent). (Note that F6 did not substantially cross the tropopause and is therefore not listed.)

| Flight No. | Date | Temp. (K) | Alt. (km) | Theta (K) | Press. (hPa) | Temp. (K) | Alt. (km) | Theta (K) | Press. (hPa) |
|---|---|---|---|---|---|---|---|---|---|
| | | Lapse rate | | | | Cold point | | | |
| F1 | 27.07.2017 | 198.2 | 15.6 | 372.7 | 109.6 | 194.6 | 17.4 | 337.7 | 82.2 |
| F2 | 29.07.2017 | 198.6 | 15.5 | 372.6 | 110.6 | 194.4 | 16.8 | 383.2 | 90.8 |
| F3 | 31.07.2017 | 192.9 | 17.0 | 386.7 | 87.7 | 192.7 | 17.0 | 388.1 | 87.7 |
| F4 | 02.08.2017 | 192.7 | 17.7 | 398.9 | 78.3 | 191.7 | 17.9 | 398.7 | 76.5 |
| F5 | 04.08.2017 | 191.5 | 15.9 | 371.8 | 104.6 | 191.5 | 16.2 | 368.0 | 99.7 |
| F7 | 08.08.2017 | 192.9 | 16.3 | 374.8 | 97.9 | 187.5 | 17.4 | 382.6 | 82.6 |
| F8 | 10.08.2017 | 194.3 | 15.7 | 366.7 | 108.3 | 186.7 | 17.1 | 375.8 | 86.7 |
| mean | | 194.4 | 16.2 | 377.6 | 99.6 | 191.3 | 17.1 | 376.3 | 86.6 |

Examples of observed temperature profiles in the Asian monsoon region versus altitude and versus potential temperature on 29 July 2017 (as well as the location of the cold point and lapse rate tropopause) are shown in Fig. 1 (top and bottom). Temperature decreases almost linearly with altitude (Fig. 1, top) up to the lapse rate tropopause and thus constitutes an example of a polytropic temperature profile (a temperature profile with a constant lapse rate; Ertel, 1938). The lapse rate tropopause is also noticeable as a discontinuity in the vertical gradient of potential temperature (Fig. 1, bottom), consistent with Kohma and Sato (2019), who defined the lapse rate tropopause by this property of potential temperature.

Note that locating the tropopause always constitutes an approximation because air parcels do not ascend vertically (as a balloon ascent does) and aircraft measurements always cover a certain horizontal area. Tropopause information at a particular point can be interpolated from gridded data, e.g. using meteorological reanalyses, but are subject to interpolation uncertainties. Ideally, the Lagrangian evolution of an air parcel before the measurement should be considered when interpreting measurements at a particular point in the lower stratosphere (e.g. Schiller et al., 2009; Wright et al., 2011; Rolf et al., 2015; Ueyama et al., 2018; Legras and Bucci, 2020; Khaykin et al., 2022; Konopka et al., 2023; Wang et al., 2025); however this view is only accessible in model simulations (see also section 3.3).

### 3.2 Cold point and lapse rate tropopause

For the first part of each StratoClim flight (e.g. the first three hours), Table 1 lists the location of the lapse rate tropopause for the same segment of the flights. The mean lapse rate tropopause is located at 16.2 km (or 377.6 K, pressure 99.6 hPa); the mean temperature at the lapse rate tropopause is 194.4 K. Similarly, for the same flight segment, Table 1, lists the cold point temperature measured by the aircraft, the altitude, the potential temperature and the pressure at the cold point. The cold point measured by the aircraft instruments for the entire flight is listed in the supplement). The range of the potential temperature for



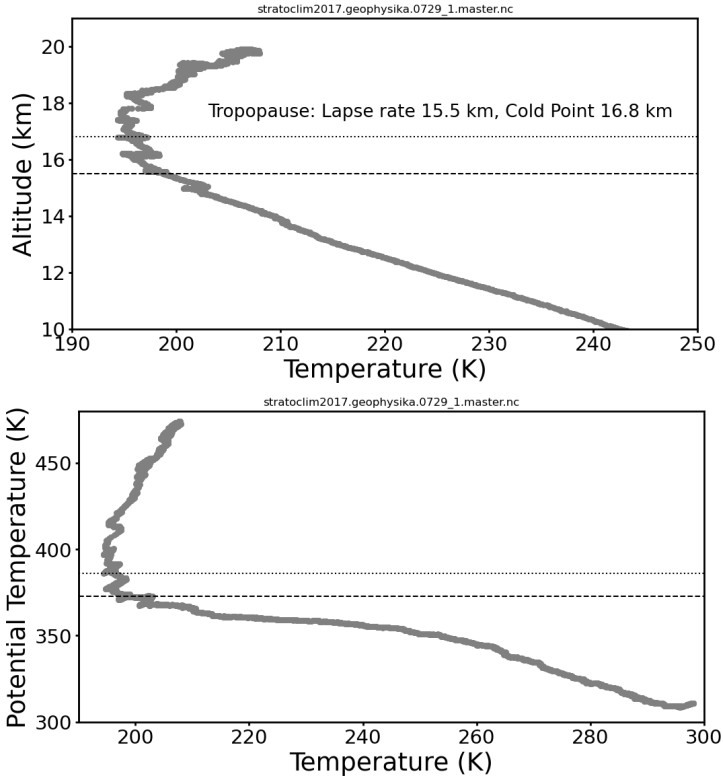

**Figure 1.** The temperature profile versus altitude (top) and potential temperature (bottom) in the vicinity of the tropopause for the scientific flight on 29 July 2017. The dashed line shows the altitude of the lapse rate tropopause (15.5 km and 372.6 K, respectively) and the dotted line the altitude of the cold point tropopause (16.8 km and (383.2 K, respectively). Tropopause and temperature information is for the first three hours of the flight (see Table 1).

the ascent flight segments at the cold point is 337.7 K to 398.7 K (approximately consistent with Khaykin et al., 2022). The mean cold point temperature for these parts of the flights is 191.3 K, the mean altitude of the cold point is 17.1 km and 376.3

K and 86.6 hPa for potential temperature and pressure, respectively. Examples of the temperature profile measured during StratoClim in the scientific flight on 29 July 2017 are shown in Fig. 1.

The observations shown here indicate that, for the Asian summer monsoon, the cold point is frequently located above the lapse rate tropopause (Table 1) although occasionally the lapse rate tropopause and the cold point may be located very closely in altitude and pressure (e.g., F5, 4 August 2017). On average the cold point is located 0.9 km above the lapse rate tropopause

and the cold point tropopause is about 3.1 K colder than the lapse rate tropopause (Table 1).

Up to the lapse rate tropopause, temperature is decreasing rapidly with potential temperature (Fig. 1. bottom) or with altitude (Fig. 1, top). Above the lapse rate tropopause, the temperature varies little up to the cold point. Above the cold point, with increasing altitude and potential temperature, temperature increases again (Fig. 1).





However, vertical ascent through layers of constant potential temperature occurs according to the heating rate per unit mass

$Q$ where the isentropic vertical velocity $d\theta/dt$ is given by the first law of thermodynamics

$$\frac{d\theta}{dt} = \frac{\theta}{T}\frac{Q}{c_p} \tag{2}$$

(here $Q/c_p$ has units of K per day). Thus, the slow vertical ascent on time scales of several days is not affected by any local transport barriers. Therefore, there cannot be a local control of the slow upward transport in the vicinity of the tropopause.

### 3.3    Ozone and Water Vapour in the vicinity of the tropopause

The chemical composition of air masses in the vicinity of the tropopause in the ASMA is investigated here using in-situ airborne measurements of ozone and water vapour in the Asian monsoon anticyclone in 2017 in the frame of the StratoClim campaign (e.g., Legras and Bucci, 2020; Krämer et al., 2020; Singer et al., 2022). Flights were selected for which both water vapour and ozone measurements are available. Throughout this work, ozone and water vapour are expressed in molar mixing ratios reported as ppb and ppm. (Note that for an ideal gas molar mixing ratios are identical to volume mixing ratios.)

The aircraft measurements discussed here cannot measure a profile in the Lagrangian sense (the Lagrangian point of view can be reached in model simulations), in other words the sampled air masses will be from downwind, i.e. from regions having in general other properties than the air masses during the ascent or descent of the aircraft (i.e., Ueyama et al., 2018; Khaykin et al., 2022; Konopka et al., 2023; Wang et al., 2025).

#### 3.3.1    Hydration and dehydration of the stratosphere through sublimating and sedimenting ice particles

The presence of ice particles in an air mass in the vicinity of the tropopause can be indicative of either hydration or dehydration. Ice particles present in subsaturated air are indicative of sublimation of ice leading to hydration of the air mass, whereas ice particles in supersaturated air will to grow and tend to sediment, thus leading to dehydration of the air mass (Jensen et al., 2020; Khaykin et al., 2022).

  The sedimentation velocity of particles of moderate size in the stratosphere is small; a particle with a radius $r = 10\,\mu$m at 30

km falls about 108 m/h and at 20 km about 90 m/h (Müller and Peter, 1992). The fall speed is $\propto r^2$ so that a smaller particle will fall substantially slower, whereas a particle with a radius $r$ of 100 $\mu$m will fall about 7 km/h at 30 km or 3 km/h at 20 km (Müller and Peter, 1992).

#### 3.3.2    The scientific flight on 29 July 2017

  The air masses sampled during the scientific flight on 29 July 2017 were influenced by convection several days *before* the

measurement (Khaykin et al., 2022; Konopka et al., 2023). But during this flight, no occurrence of particles was observed above 13 km and in particular above the lapse rate tropopause (Fig. 2, bottom). There are two layers of enhanced water vapour of about 10 ppm between 16.5 and 17.5 km, at and above the cold point tropopause (Fig. 2, top right). The convective origin of these layers of enhanced water vapour at about 16.5 and 17.5 km is confirmed by a strong enhancement in the HDO/H$_2$O ratio (see also section 2.3) indicating that the water vapour enhancement was produced by sublimation of ice (Khaykin et al.,




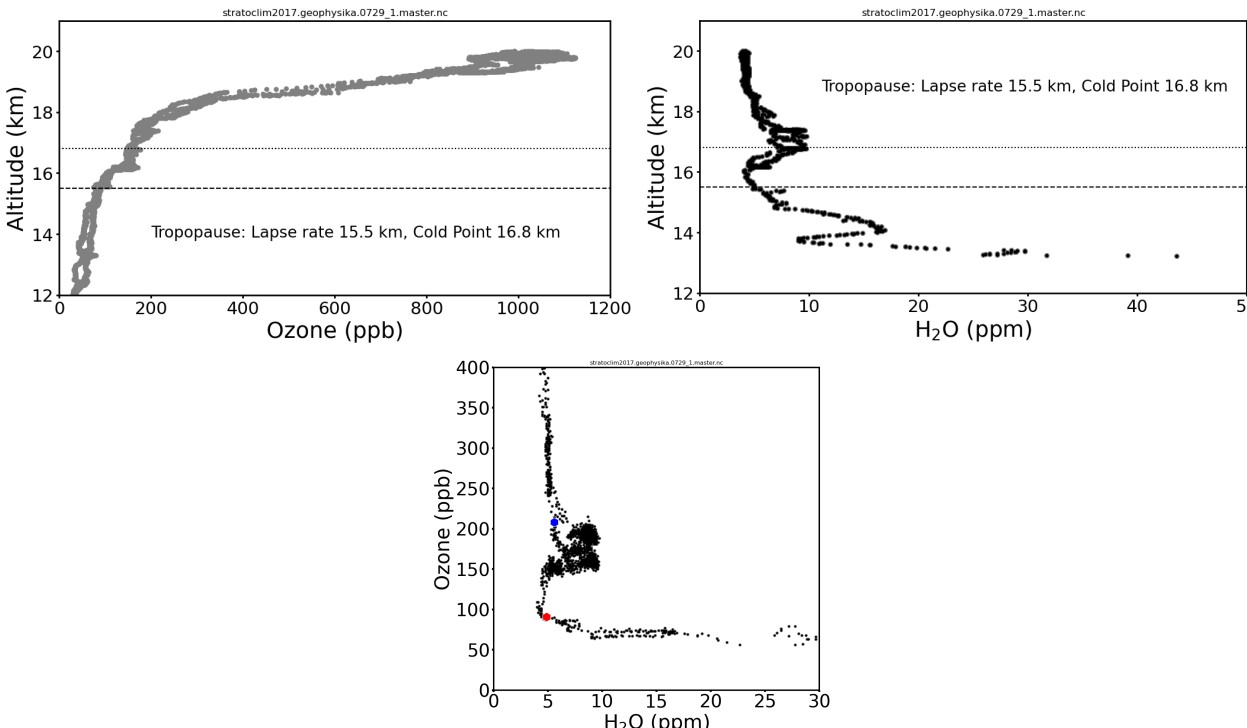

**Figure 2.** Measurements of molar mixing ratios of ozone (top left) and water vapour (top right) versus altitude in the vicinity of the tropopause for the scientific flight on 29 July 2017. The dashed line shows the altitude of the lapse rate tropopause and the dotted line the altitude of the cold point tropopause. Bottom panel shows a scatter plot of the aircraft measurements of ozone and water vapour on 29 July 2017. The red symbol indicates location of the lapse rate tropopause; the blue symbol location of the cold point tropopause. Tropopause information is for the first three hours of the flight (see Table 1), ozone and water vapour measurements are for the entire flight. In the top right and bottom panel, grey symbols indicate the FISH total water vapour measurements and black points show the FLASH gas-phase water vapour measurements. Note that for this flight the FISH and FLASH water vapour measurements are very similar (top right and bottom panel), in other words no cloud particles were observed.




2022). On this day, water vapour measurements above the cold point tropopause show a decline of $H_2O$ with altitude, with no indication of dehydration above the cold point (Fig. 2).

Ozone mixing ratios stay approximately constant with altitude up to about 12 km, with a moderate increase above. A substantial ozone increase starts above the lapse rate tropopause, increasing substantially with altitude, especially above the cold point (Fig. 2).

These findings are corroborated by investigating the ozone-$H_2O$ scatter plot (Fig. 2, bottom); which shows the well known "L-shape" (Pan et al., 2018) and, further, the water vapour enhancements of about 10 ppm for ozone mixing ratios between $\sim$ 150 to 200 ppb. Measurements below the lapse rate tropopause are of tropospheric character (poor in ozone and enhanced in water vapour).

### 3.3.3 The scientific flight on 8 August 2017

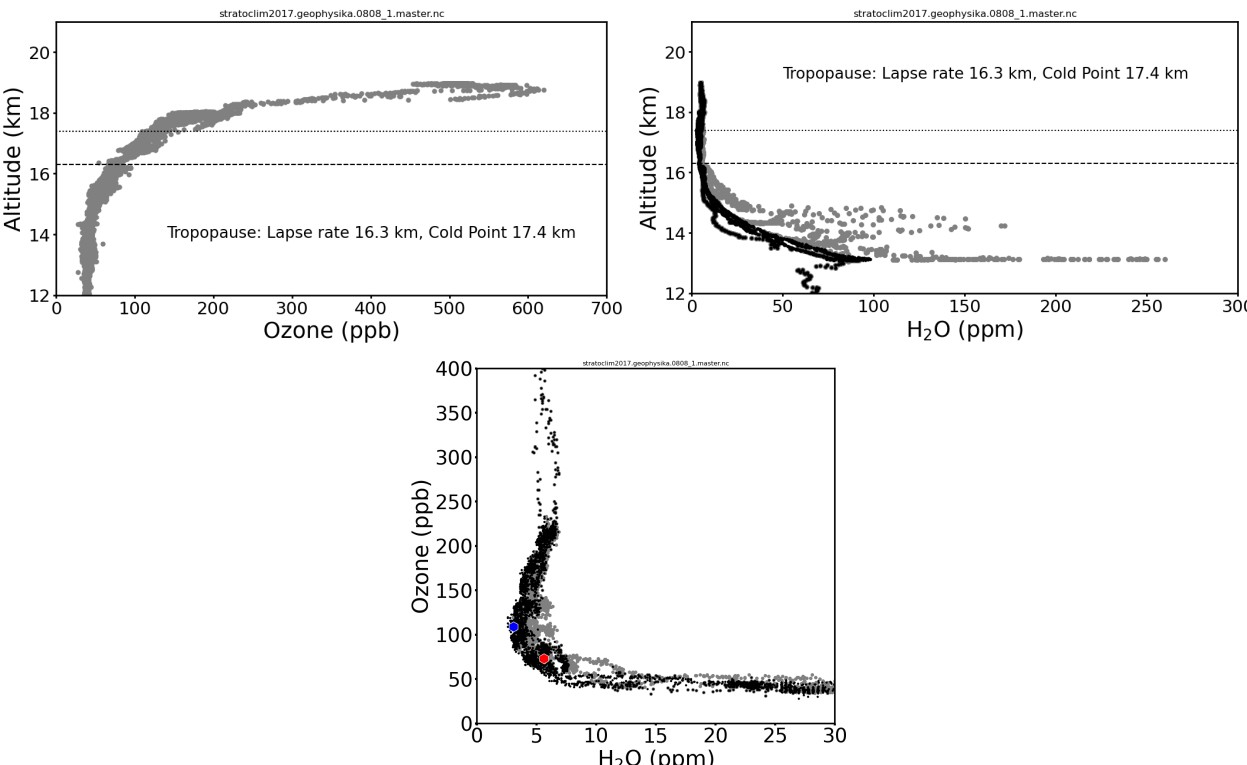

**Figure 3.** Measurements of ozone (top left) and water vapour (top right) versus altitude in the vicinity of the tropopause for the scientific flight on 8 August 2017. The dashed line shows the altitude of the lapse rate tropopause and the dotted line the altitude of the cold point tropopause. Tropopause information is for the first three hours of the flight (see Table 1), ozone and water vapour measurements are for the entire flight. In the bottom panel, grey symbols indicate the FISH total water vapour measurements and black points show the gas-phase water vapour measurements by the FLASH instrument.




During the flight on 8 August 2017, air masses were sampled that show an approximately constant ozone mixing ratio with altitude up to 14 km; the ozone mixing ratio increases somewhat above. Ozone mixing ratios increase substantially above the lapse rate tropopause (Fig. 3, top left panel).

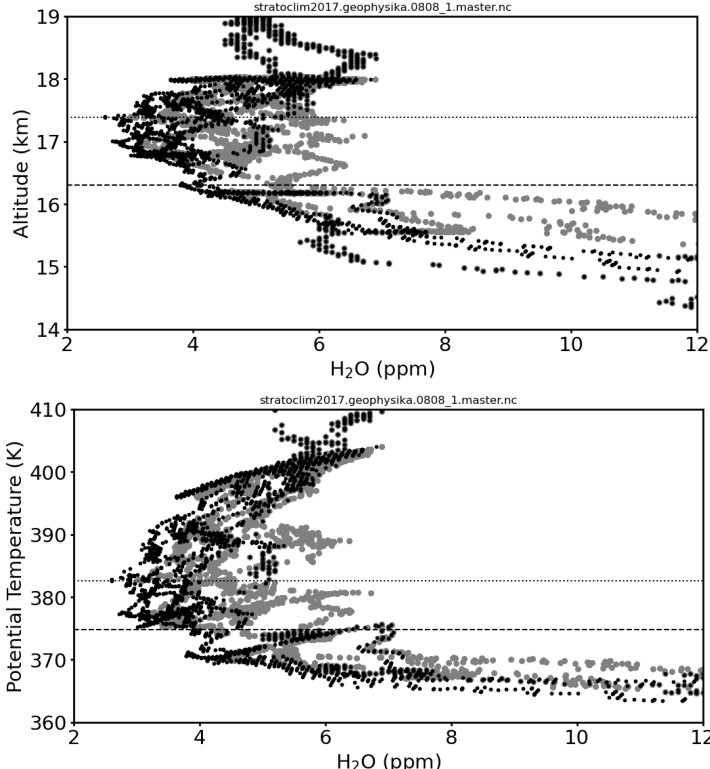

**Figure 4.** Water vapour versus altitude (top) and versus potential temperature (bottom) in the vicinity of the tropopause for the scientific flight on 8 August 2017 (similar as in Fig. 3, bottom), but for a reduced range of observed water vapour values. The dotted line shows the cold point for this flight (17.4 km, top and 382.6 K, bottom) and the dashed line the lapse rate tropopause (16.3 km, top and 374.8 K, bottom). Grey symbols indicate the FISH total water vapour measurements and black points show the gas-phase water vapour measurements by the FLASH instrument. The aircraft measurements extend to about 440 K.

Below the lapse rate tropopause, on this day, ice particles occur (Fig. 3, top right panel, Fig. 4). This is obvious as the grey symbols (total water) clearly show – at the same altitude or potential temperature – higher values than the black symbols

(gas-phase water vapour). The same is true (but with lower values of total water) for the vertical range between the lapse rate tropopause and the cold point tropopause (Fig. 4). Slightly above the cold point tropopause, at about 390 K (17 km), ice particles are observed (Fig. 4), but no clear indication of particle occurrence is found above 390 K. The measurements show enhancement of water vapour to mixing ratios up to $\sim$ 7 ppm (Fig. 4) above the cold point tropopause between 400 and 410 K.





The ozone-$H_2O$ scatterplot for 8 August (Fig. 3, bottom panel) corroborates the findings deduced from the observed profiles.

The air below the lapse rate tropopause is of tropospheric character. Particles occur for ozone mixing ratios of 100-150 ppb and the $H_2O$ enhancement with mixing ratios below about 7 ppm is noticeable for ozone mixing ratios of 200-350 ppb.

### 3.3.4    The scientific flight on 10 August 2017

The scientific flight on 10 August 2017 was conducted under conditions of strong convection (Konopka et al., 2023) for about the first two hours of the flight. The entire flight duration was 3.7 hours.  Clearly noticeable in the water vapour measurements

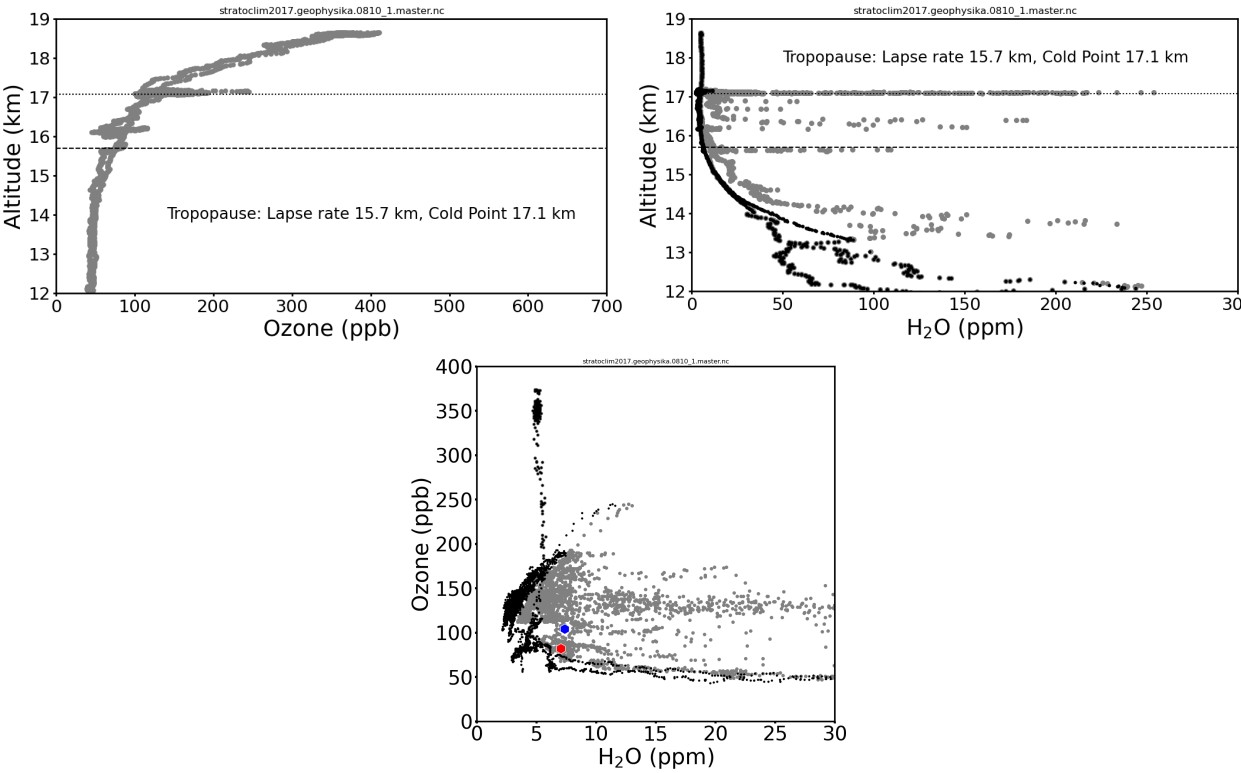

**Figure 5.** Measurements of ozone (top left) and water vapour (top right) versus altitude in the vicinity of the tropopause for the scientific flight on 10 August 2017. The dashed line shows the altitude of the lapse rate tropopause and the dotted line the altitude of the cold point tropopause. Tropopause information is for the first three hours of the flight (see Table 1), ozone and water vapour measurements are for the entire flight. In the bottom panel, grey symbols indicate the FISH total water vapour measurements and black points show the gas-phase water vapour measurements by the FLASH instrument.

on 10 August (Fig. 5, top right) is the very strong total water enhancement, which indicates the occurrence of ice particles (enhanced FISH water vapour measurements reaching up to $\sim$ 250 ppm). These ice particles are closely collocated with the altitude of the cold point tropopause (Fig. 5, top right and Fig. 6, top). There is also occurrence of particles below the cold point as well as at and below the lapse rate tropopause (Fig. 5, top right). Further, cloud particles (i.e., high total water values)



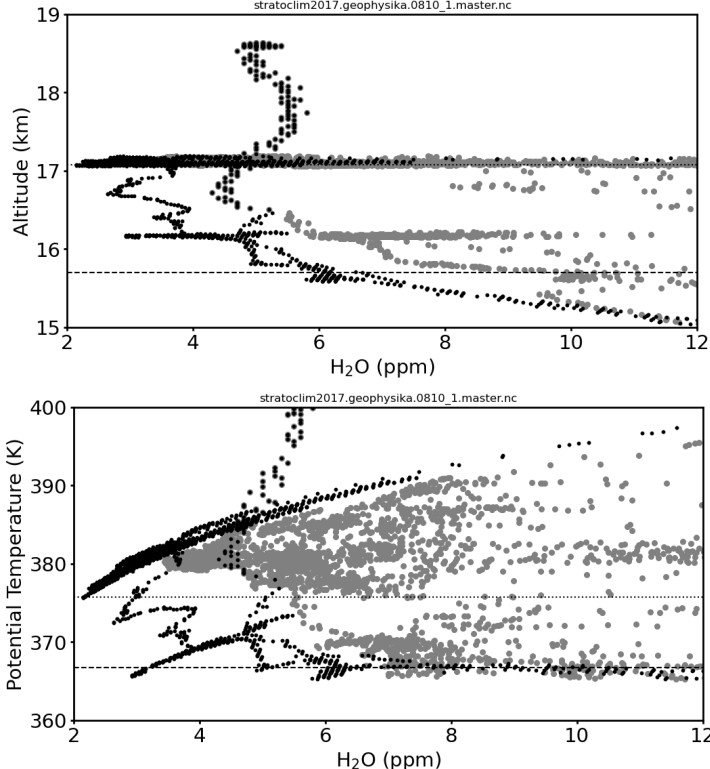

**Figure 6.** Water vapour versus altitude (top) and versus potential temperature (bottom) in the vicinity of the cold point and the lapse rate tropopause for the scientific flight on 10 August 2017 (similar as in Fig. 5, bottom, but for a reduced range of observed water vapour values). The dotted line shows the cold point for this flight (17.1 km, top and 375.8 K, bottom) and the dashed line the lapse rate tropopause (15.7 km, top and 366.7 K, bottom). Grey symbols indicate the FISH total water vapour measurements and black points show the gas-phase water vapour measurements by the FLASH instrument. The aircraft measurements extend to about 420 K.

occur below the lapse rate tropopause in the troposphere. Ice particles are observed during the first two hours of the flight. No

ice particles were detected by FISH for the flight section between about 17.1 km (375 K) and the top altitude of the flight ($\sim$ 18.5 km, 420 K). This, the flight section between $\sim$ 17.1 km and the top altitude of the flight was conducted in particle free air (Fig. 6).

The part of the flight on 10 August 2017 between 9:45 and 11:10 was conducted close to the pressure level of the cold point tropopause (not shown). Thus, the particle occurrence (i.e., the enhanced water vapour measured by FISH) close to the cold

point tropopause (Fig. 5, top right panel) consists of a series of peaks of enhanced water vapour sampled between 9:45 and 11:10.

Ozone mixing ratios on 10 August 2017 are low and not very variable between 12 and 14 km ; they increase slightly between 14 km and the lapse rate tropopause (Fig. 5, top left panel). Ozone mixing ratios increase above the lapse rate tropopause and




are much more variable. Ozone increases further above the cold point tropopause and there is a large variability of ozone at the
cold point tropopause, with mixing ratios ranging between ∼ 100 and 350 ppb.

The ozone-$H_2O$ scatterplot for 10 August 2017 (see Fig. 5, bottom) corroborates the findings deduced from the observed
profiles (Fig. 5, top); the scatterplot shows in general an "L-shape" with an enhancement of gas-phase water vapour at about
60-170 ppb ozone. Very low water vapour mixing ratios (below about 3 ppm) are observed during this flight (for ozone mixing
ratios of about 100-150 ppb).

These water vapour observations indicate that (under the conditions of strong convection) dehydration occurs at altitudes
close to the cold point tropopause (Figs. 5, top right and 6), where there is supersaturation (Fig. 7). However, using potential
temperature as the vertical coordinate (Fig. 6, bottom) indicates that particle occurrence under conditions of strong convection
extends to about 390 K.

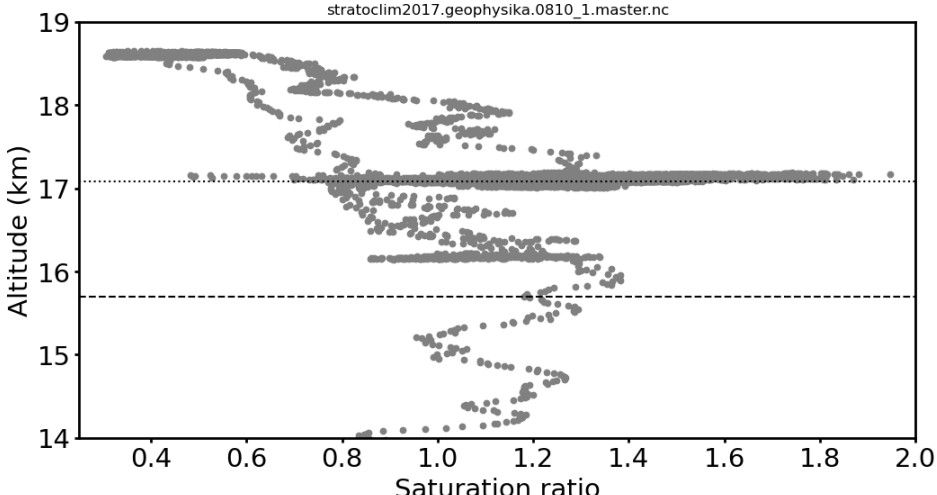

**Figure 7.** Saturation ratio of $H_2O$ over ice versus altitude for the scientific flight on 10 August 2017 in the vicinity of the cold point and the
lapse rate tropopause. Gas-phase $H_2O$ measurements are from the FLASH instrument; these measurements are used to calculate the water
vapour saturation ratio. The dotted line shows the cold point for this flight (17.1 km) and the dashed line the lapse rate tropopause (15.7 km).
The top altitude for which FISH observed ice particles is 17.1 km. (See supplement for a figure showing potential temperature as the vertical
axis.)

Importantly, the particles for conditions of strong convection (the first two hours of the flight) occur in strongly supersaturated
air (Fig. 7). Thus these particles will grow and therefore likely sediment. Sedimenting ice particles will dehydrate the air mass
they occur in; thus for the flight on 10 August, supersaturated air in the vicinity of the cold point should lead to dehydrating
the air mass at the cold point tropopause.

However, particles and supersaturated air masses occur between the cold point and the lapse rate tropopause, indicating
further dehydration occurring at these altitudes. Particles also occur in the troposphere, below the lapse rate tropopause (Fig. 5,
top right).





### 3.3.5 General properties of the cold point and the lapse rate tropopause in the Asian summer monsoon anticyclone

Up to the lapse rate tropopause, ozone mixing ratios are relatively low, the air is humid and frequent occurrence of cloud particles is observed. Water vapour mixing ratios decrease with increasing altitude and with slightly increasing ozone mixing ratios. Thus we conclude that the lapse rate tropopause in the Asian summer monsoon anticyclone constitutes a good estimate
for the upper boundary of the well mixed tropospheric air mass.

Between the lapse rate tropopause and the cold point, the air masses show no longer a tropospheric character, ozone mixing ratios increase with altitude. Further, convection may penetrate the lapse rate tropopause and humidify the region above the lapse rate tropopause. Water vapour mixing ratios in the ASMA in 2017 above the cold point tropopause (i.e., also above the lapse rate tropopause) range up to $\sim$ 5-6 ppm. Moister patches with enhancements in water vapour mixing ratios up to $\sim$
10 ppm are occasionally observed. However, for strong convection, severe dehydration is indicated by the occurrence of ice particles in supersaturated air very close to the cold point tropopause.

## 4   Discussion

Here, with a focus on the Asian Monsoon, we confirm earlier measurement reports for the tropical west Pacific in winter (Pan et al., 2018) that the lapse rate tropopause identifies the transition from tropospheric air masses to stratospheric air masses.
We also find that in the Asian monsoon the air is generally drier at and above the cold point tropopause than below; again consistent with earlier reports for the west Pacific (Pan et al., 2018). However, the air above the cold point tropopause in the Asian summer monsoon is more humid (mixing ratios of $\sim$ 3-10 ppm) than in the west Pacific in winter ($\sim$ 3 ppm, Pan et al., 2018).

Ozone mixing ratios in the ASMA, increase moderately with altitude in the upper troposphere (below the lapse rate tropopause)
and increase substantially above the lapse rate tropopause (e.g., Pan et al., 2018; Brunamonti et al., 2018; Fadnavis et al., 2023). Ozone mixing ratios between the cold point tropopause and the lapse rate tropopause range between 50 and 200 ppb, consistent with measurements in the west Pacific in winter (Pan et al., 2018).

In the Asian monsoon region, penetration of deep convective systems to above the lapse rate tropopause (Clapp et al., 2023) and also (but to a lesser extent) above the cold point tropopause occurs. If ice particles reach the lowermost stratosphere and
sublimate there (before the particles sediment) this process may cause local hydration. Indeed, layers of enhanced gas-phase water vapour (below $\approx$ 10 ppm) above the lapse rate and the cold point tropopause were observed on 29 July and 8 August 2017 (Lee et al., 2019; Khaykin et al., 2022, see also Figs. 2, 3 and 6). There are also occasions of observations of enhanced water vapour mixing ratios of $\approx$ 10 ppm in the lowermost stratosphere over North America during and before the American monsoon (Robrecht et al., 2019). However, such conditions of enhanced water vapour are not observed frequently.
On the flight on 8 August 2017 (Fig. 4), the occurrence of ice particles was observed at $\sim$ 390 K (17 km) and an enhancement in gas-phase water vapour at $\sim$ 400-410 K (18-19 km) above the cold point tropopause in agreement with earlier reports (Lee et al., 2019; Khaykin et al., 2022). In moist, supersaturated, air parcels, prevailing above the cold point tropopause, formation of ice particles is possible. However, it is not obvious that these ice clouds cause substantial dehydration because of a limited





spatial and temporal extent of such air parcels and because of a limited size (causing a low sedimentation velocity) of the ice
particles (Khaykin et al., 2022).

Overall, we observe a certain variability of gas-phase water vapour in the Asian summer monsoon region as both dehydration
of air caused by freeze drying (water vapour mixing ratios below about 3 ppm) and hydration caused by convective overshoots
(water vapour mixing ratios $\lesssim$ 10 ppm) of the cold point tropopause occurs (consistent with the findings of Emmanuel et al.,
2021).

The scientific flight on 10 August 2017 allowed the tropopause conditions of strong convection to be investigated. Kuang
and Bretherton (2004) report that the cold point tropopause under such conditions is strongly tied to the convective cooling
maximum. This is consistent with substantial ice particle occurrence around the cold point tropopause as observed during the
flight on 10 August 2017 (Figs. 5 and 6). The ice particles at the cold point exist in strongly supersaturated air (Fig. 7) and will
thus likely sediment. This would lead to dehydration at the cold point tropopause until a saturation of one is reached, again
consistent with the conclusions of Kuang and Bretherton (2004). The findings by Kuang and Bretherton (2004) that the cold
point tropopause is strongly tied to convection suggests that dehydration processes at the cold point can be assumed as abrupt.

Our analysis for the flight on 10 August indicates that stratospheric water vapour in the Asian summer monsoon region is
controlled by cold point temperatures, and that deep convection overshooting the lapse rate tropopause plays a relatively minor
role in moistening the stratosphere (supporting conclusions in earlier work based on remote sensing observations, Randel et al.,
2015). Further, the conclusion of very rapid transport from the ground up to the cold point on 10 August 2017 is also supported
by the $CO_2$ measurements during this flight: $CO_2$ mixing ratios in the vicinity of the cold point (about 375 K, Fig. 6) are of
lower tropospheric character (mixing ratios below about 400 ppm, Vogel et al., 2024, Fig. A3).

## 5   Conclusions

The lapse rate tropopause, in the ASMA, marks the top boundary of tropospheric air. Tropospheric air is vertically mixed,
humid and relatively low in ozone; insofar, the lapse rate tropopause in the Asian summer monsoon constitutes an air mass
boundary. Further, the lapse rate tropopause (in contrast to the cold point tropopause) constitutes a clear discontinuity in the
profile of temperature against potential temperature (Fig. 1, bottom and electronic supplement). Nonetheless, the demarcation
between the troposphere and the stratosphere is different for different trace species, for example there is freeze out of water
vapour at the lowest temperatures, chemical production of ozone in the stratosphere and limited inmixing of aged lower
stratospheric air into tropospheric air ascending into the lowermost stratosphere. Vertical ascent ($\dot{\theta}$) in the ASMA through
layers of constant potential temperature $\theta$ occurs according to the atmospheric heating rate (Eq. 2) with no local (long-term)
obstacle for vertical transport. The location of the lapse rate tropopause in the Asian summer monsoon is thus *not* controlled
by local processes.

Thus, the air masses below and above the lapse rate tropopause are very different. In other words, the lapse rate tropopause
and the cold point tropopause are two different things (in spite of both being referred to as "tropopause"); they constitute bound-
aries of different air masses. Commonly, the cold point is located substantially (about 1 km) above the lapse rate tropopause.



Above the cold point tropopause in the Asian Monsoon, the air is largely stratospheric; i.e., the air is dry and ozone mixing ratios increase with altitude.

Therefore, it is necessary to specify if convective systems are *overshooting* the lapse rate or the cold point tropopause. On some occasions, deep convective systems may penetrate the cold point tropopause. Under such conditions, sublimating cloud particles can humidify the air above the cold point tropopause. Such conditions were observed during the flight on 8 August 2017 (Fig. 4) with total water mixing ratios of $\lesssim 7$ ppm $\sim 1$ km ($\sim 25$ K) above the cold point tropopause.

Moist plumes caused by convection above the cold point tropopause may reach a layer characterised by radiatively driven upward motion of $\approx 1$ K/d (Legras and Bucci, 2020). Thus, such moist air parcels – as long as they remain unmixed – will rise to greater potential temperatures (altitudes). This type of air parcels was also detected in balloon measurements in the ASMA above the cold point tropopause in 2016 and 2017 (Brunamonti et al., 2018). However, a hydration of the lowermost stratosphere is not regularly observed. Further, in such moist air parcels, new ice cloud formation is possible, but substantial dehydration is unlikely (Khaykin et al., 2022).

When convection is strong (flight on 10 August 2017), particle occurrence and oversaturation is observed in close proximity of the lapse rate tropopause (Fig. 5, top right, Fig. 6 and Fig. 7). A very strong enhancement of total water (i.e., ice particle occurrence) is noticeable close to the cold point tropopause at about 17.1 km (about $\pm 100$ m around the cold point, Fig. 6). The coincidence of the cold point tropopause and the ice particle occurrence at an altitude of $\sim 17.1$ km leads to the hypothesis that the same processes are responsible for the formation of the cold point and for the formation of cloud particles. Thus strong convection should lead to dehydration at the cold point tropopause and not to a moistening of the stratosphere.

Further, a clear enhancement of ozone mixing ratios (up to about 250 ppb) is visible at the location of the cold point tropopause (Fig. 5). There is no indication in the Geophysica measurements in the Asian summer monsoon in 2017 of humidification of stratospheric air exceeding mixing ratios of $\sim 10$ ppm of water vapour above the cold point tropopause. Overall, the air in the lower stratosphere above the Asian monsoon anticyclone is moister (water vapour mixing ratios of 3-10 ppm) than in winter over the west Pacific (water vapour mixing ratios of $\sim 3$ ppm).

Measurements of the species and quantities considered here (namely ozone, water vapour, temperature and cloud occurrence) are in principle also accessible through balloon-borne measurements (e.g., Bian et al., 2012; Renard et al., 2016; Gao et al., 2016; Brunamonti et al., 2018; Bian et al., 2020; Hanumanthu et al., 2020; Fadnavis et al., 2023; Clemens et al., 2024). Measurements of cloud occurrence are not straightforward as it is difficult to cover the entire size range of cloud particles occurring in the atmosphere in measurements. In the future, when fewer satellite observations will be available (Salawitch et al., 2025) networks of balloon-borne observations might become increasingly important (Müller et al., 2016; Bian et al., 2020; Fadnavis et al., 2023). Therefore, in future work, an extension of the present analysis for the Asian monsoon will be possible (based on balloon-borne measurements), which will feature more profiles (and thus a better statistics) and will allow inter-annual as well as intra-seasonal variability to be addressed.





**Appendix A: The relation between altitude, potential temperature and pressure in the Asian summer monsoon**

Frequently, there is the question regarding the relation between altitude, potential temperature and pressure in the upper tropo-
sphere and lower stratosphere (Krämer et al., 2020, Fig. 2) and in particular in the ASMA. Therefore, empirical fits (based on
the aircraft measurements in the Asian summer monsoon in 2017) for the relations in question are provided here. Altitude is
taken from the aircraft system (UCSE, Stefanutti et al., 1999). The empirical (polynominal) fits are given below, the coefficients
used for the fits are listed in Table A1. For altitude and potential temperature one may use the ansatz:

$$\text{Altitude(km)} = a \cdot x^4 + b \cdot x^3 + c \cdot x^2 + d \cdot x + e \tag{A1}$$

and for pressure and potential temperature

$$\text{Pressure(hPa)} = a \cdot x^5 + b \cdot x^4 + c \cdot x^3 + d \cdot x^2 + e \cdot x + f \quad, \tag{A2}$$

where $x$ is potential temperature (see also Figs. A1 and A2).

The relation between pressure $p$ and altitude $z$ (for an altitude range 8-20 km) is described well by an exponential dependence
(not shown)

$$p = p_{\text{fit}} \cdot e^{(-z/H_{\text{fit}})} \tag{A3}$$

with $p_{\text{fit}} = 1242$ hPa and $H_{\text{fit}} = 6.42$ km and where there is little variation of the relation above and below the tropopause (note
that the values $p_{\text{fit}}$ and $H_{\text{fit}}$ are only fitting parameters, the relation is valid for 8-20 km). The approximate values derived from
Eq. A3 are shown in table A2.

**Table A1.** Parameters for the empirical fits

| Parameter | Value (altitude fit) | Value (pressure fit) |
|---|---|---|
| $a$ | $-2.25014801 \cdot 10^{-7}$ | $-9.09161964 \cdot 10^{-8}$ |
| $b$ | $3.87672005 \cdot 10^{-4}$ | $1.97741637 \cdot 10^{-4}$ |
| $c$ | $-2.50052963 \cdot 10^{-1}$ | $-1.71667004 \cdot 10^{-1}$ |
| $d$ | $7.15952467 \cdot 10^{+1}$ | $7.43565770 \cdot 10^{+1}$ |
| $e$ | $-7.66247724 \cdot 10^{+3}$ | $-1.60697605 \cdot 10^{+4}$ |
| $f$ | – | $1.38641595 \cdot 10^{+6}$ |





**Table A2.** Approximate pressure (Eq. A3) and potential temperature (Eq. A1) versus altitude for the Geophysica flights in 2017.

| Alti. (km) | Press. (hPa) | Theta (K) |
|---|---|---|
| 5. | 570 | 342 |
| 6. | 487 | 344 |
| 7. | 417 | 346 |
| 8. | 357 | 348 |
| 9. | 305 | 350 |
| 10. | 261 | 352 |
| 11. | 224 | 355 |
| 12. | 191 | 358 |
| 13. | 164 | 361 |
| 14. | 140 | 365 |
| 15. | 120 | 369 |
| 16. | 103 | 376 |
| 17. | 88 | 385 |
| 18. | 75 | 409 |
| 19. | 64 | 444 |
| 20. | 55 | 480 |

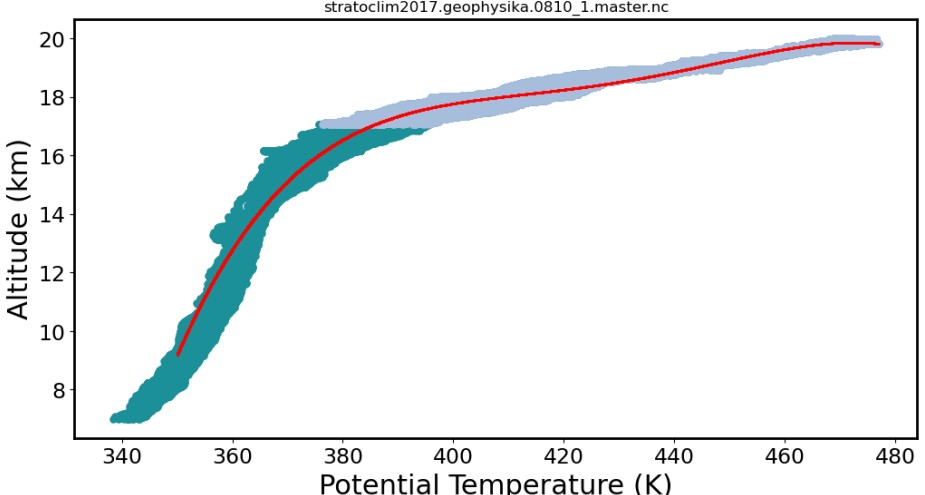

**Figure A1.** The relation between altitude and potential temperature for all science flights from Kathmandu (dark). Measurements above the (mean) cold point tropopause are shown in light blue. Red line shows an empirical fit to the data valid for potential temperature greater than 350 K.



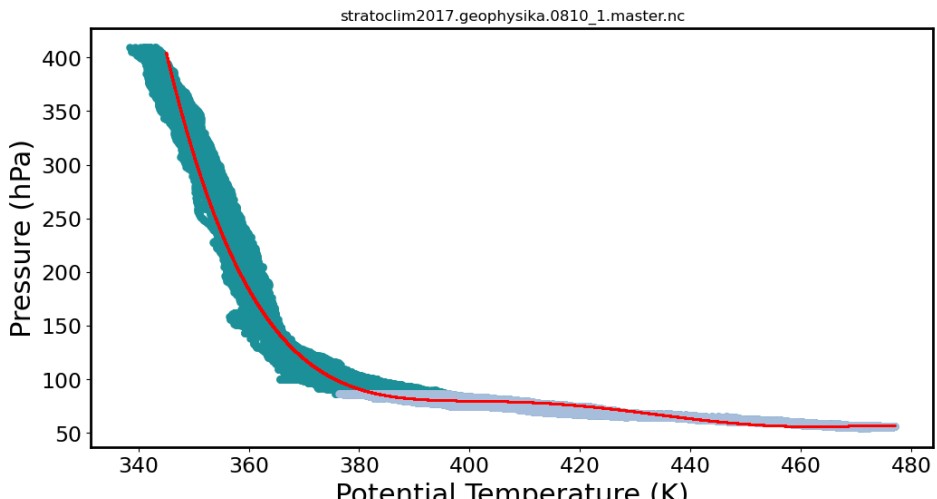

**Figure A2.** The relation between pressure and potential temperature for all science flights from Kathmandu (dark). Measurements above the (mean) cold point tropopause are shown in light blue. Red line shows an empirical fit to the data for potential temperature greater than 345 K.



*Code availability.* The code to calculate the location of the lapse rate tropopause is available at https://gitlab.physik.uni-muenchen.de/LDAP_ag-birner/tropopause/-/blob/master/tropopause.py.

*Data availability.* The measurements used here were obtained during the scientific flights of the StratoClim aircraft campaign in summer 2017; the data are available on the HALO database at https: //halo-db.pa.op.dlr.de/mission/101 (DLR, 2025).

*Author contributions.* M.K., C.R. and N.S. provided the FISH measurements used here; F.R. provided the ozone measurements used here.
All co-authors discussed the results of the study and helped formulating the manuscript.

*Competing interests.* Some of the authors are editors of ACP; otherwise the authors declare that they have no competing interests

*Acknowledgements.* We thank Fred Stroh very much for help with data processing and for comments on the quality of the measurements used here. We further thank Matthias Riße and Gebhard Günther very much for help with the python code used for this study. We thank the TDC and the UCSE team very much for producing thermodynamic measurements, A. Lykov, V. Yushkov and the FLASH team for
the gas-phase water vapour measurements, A. Ulanovsky, V. Yushkov and the FOZAN team for measurements of ozone on the high-flying laboratory Geophysica. We thank P. Konopka for helpful discussions.



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
