# Peer review of "The lapse rate cold point tropopause in the Asian Summer Monsoon anticyclone"

_EGUsphere, 2025_

## Referee Comment (RC1)

**Review of egusphere-2025-5761, Müller et al., The lapse rate.........**

GENERAL

It is pleasing to see this unique study of strat-trop exchange in connection with monsoonal circulations. The work is of a high standard, original and is clearly suitable for publication in *Atmos Chem Phys.* In the COMMENTARY below I make some points that may be helpful. In particular, the references are long on recent work and somewhat short on original contributions.

COMMENTARY

lines 35 et seq: This section misses out some earlier references that deal with research quality observations, and which say some relevant things about the tropical tropopause [1,2,3,4,5]. In particular, the average $\theta_w$ at the sea surface in the tropics is 355 K, significantly lower than the potential temperature of the tropical tropopause, however that is defined. Transport from the midlatitude stratosphere and local descent must contribute [3]. Reference [5] below has some highly relevant discussion, also applicable in large parts of the subsequent text. The original point about the role of the monsoon was made in [6], amplified in [7]. They should be referenced.

lines 45-50: It would be helpful to the reader if the location of the StratoClim campaign was supplied.

line 69: In 1972 the NASA WB57F flew around a Cb top and anvil penetrating the tropopause near Amarillo, Texas. It saw higher concentrations of water vapour downwind than upwind of the penetrating top and anvil. It is reported in the CIAP report, published by the US Dept of Transportation in 1974.

lines 73-79: These are all model based arguments. Observational ones, considerably better founded, are in [5] and [8], in which the TTL (not then called that) was pointed out.

line 118: Saturation over ice is a crude approximation, see discussion in [5].

lines 133-140: Isotopic studies are very useful in principle. One process that needs discussion is the very fast exchange rates at aqueous particle surfaces. For example, an O atom in a water molecule that enters a particle may not be attached to the same H (and D) atoms when it leaves. Particles are not at equilibrium; Henry's Law does not apply.

lines 163-166: Such trajectories are Lagrangian sampling of an Eulerian system, so are not truly following a given unmixed air mass.

lines 200-207: See the discussion in [9].as referenced in [5].

lines 272-281: Interesting. In general, for the whole paper, I find it surprising that there is no mention of the subtropical jet stream and its migration north during boreal summer. See [10].

lines 283-288: Note that an early stratosphere-troposphere GCM predicted the role of monsoonal circulation in air entry to the stratosphere [12].

lines 318 & 332: "Control" is a slippery concept, given the nonconvergent variance of airborne observations [11].

lines 334-338: See Figure 14 in [3].

[1] *Q J R Meteorol Soc, 122,* 929-944 (1996)

[2] *Q J R Meteorol Soc,* **124,** 1559-1578 (1998)

[3] *J Geophys Res,* **108(D23),** 4734 (2003)

[4] *J Geophys Res,* **109,** D05310 (2004)

[5] *J Geophys Res,* **111,** D13304 (2006)

[6] *J Geophys Res,* **102,** 13213-13234 (1997)

[7] *Q J R Meteorol Soc,* **125,** 1079-1106 (1999)

[8] *Q J R Meteorol Soc,* **123,** 1-69 (1997)

[9] *J Geophys Res,* **98,** 8639-8664 (1993)

[10] *Q J R Meteorol Soc,* **106,** 227-253 (1980)

[11] *Entropy,* **27(7),** 740 (2025)

[12] *Q J R Meteorol Soc,* **110,** 321-356; 357-392 (1984)

---

## Referee Comment (RC2)

The manuscript entitled 'The lapse rate cold point tropopause in the Asian Summer Monsoon Anticyclone' uses high altitude aircraft measurements from the StratoClim campaign in 2017 to explore the lapse rate and cold point tropopause characteristics in the Asian summer monsoon anticyclone (ASMA). This manuscript uses water vapour, ozone, and temperature measurements from multiple research flights, including convective flights. They find the lapse rate tropopause to better distinguish between the tropospheric and stratospheric air masses. In convective cases, there is hydration near and above the lapse rate tropopause. In strong convective cases, dehydration occurs near the cold point tropopause. I found this to be an interesting study, and section 3.3.5 to be particularly insightful. My main comments are bulleted below, and I recommend minor revisions prior to publication.

**Minor Comments**
*Abstract*
In the abstract, L3-4 state that this study finds the lapse rate tropopause, rather than the cold point, to be a good estimate of the upper boundary of the troposphere. However, the remainder of the abstract focuses on the cold point when referencing water vapour mixing ratios, ozone mixing ratios, as well as hydration and dehydration patches. If the lapse rate tropopause better serves as the boundary, why is there more focus given to the cold point?

*Introduction*
L25: The impact to the radiation budget is an important part of the transport of water vapour to the lower stratosphere in the referenced papers, it could strengthen this point to mention that specifically.

L95-98: Could you include a reference for the differences between the lapse rate and cold point tropopause regarding water vapour in the ASMA?

*Methods*
It would be beneficial to have further information about the StratoClim campaign here. Including some details such as where the campaign took place, what aircraft was used, how many flights total versus what will be examined here, and what the objective were of the flights analyzed here would all provide helpful context to the reader.

L101: Please specify what the "research aircraft" is.

L137: Omit the "." in "particles. (Khaykin et al., 2022) are…".

*Results*
L176, L179: Including some additional, but brief, context about the mentioned flights (ex: convective or not) would be helpful to the reader when certain flights are mentioned but not the main flight of the sub-discussion (ex: 4 August flight) to know why certain flights are focused on and what attributes may be similar to flights that receive more discussion.

L215: More clearly distinguishing why the authors choose to focus on the cold point tropopause for water vapour in the results after stating that the lapse rate tropopause better separates between the troposphere and stratosphere would help emphasize the novelty of the results here. I find it can be less clear why the focus switches between tropopause definitions for different results discussion otherwise (similar to abstract comment above).

L225: This subsection is focused on a different flight, what is the nature of the 8 August 2017 flight? Is it similar to the 29 July 2017 flight?

L228: Is "this day" referencing the 8 August 2017 flight?

L252: Typo with "14 km ; they".

*Conclusions*
L360: The discussion of future work in this paragraph is nice; it would be beneficial to add some mention of limitations to this work within the Discussion or Conclusions sections as well.

*Figures*
All Figures: Each figure has a .nc file name above it as a title. These titles do not seem to be needed, and the small text size makes it challenging to read. I recommend omitting these titles and replacing them with a descriptive title or omitting the titles altogether.

Figure 1: Typo in caption with extra parenthesis in "(16.8 km and (383.2 K, respectively)."

Figures 3,5: What do the red and blue dots represent?

Figure 5: What is grey versus black for the upper right plot?

---

## Author Comment (AC1)

**Reply to review comments on:**

**"The lapse rate and the cold point tropopause in the Asian Summer Monsoon anticyclone"**

**Reply to Review 1**

We thank the reviewer very much for reading the paper and for valuable comments that helped improving the paper. The reported literature (in particular older literature) was particularly important. All the comments are repeated below and the corresponding changes to the manuscript are reported also. The comments by the reviewer are repeated in the reply in *italics* and the response by the authors are in roman font. The references given in the comments are given in square brackets as in the review.

*It is pleasing to see this unique study of strat-trop exchange in connection with monsoonal circulations. The work is of a high standard, original and is clearly suitable for publication in* Atmos Chem Phys. *In the COMMENTARY below I make some points that may be helpful. In particular, the references are long on recent work and somewhat short on original contributions.*

We thank the reviewer for these positive statements. Regarding the references we have followed the advice and included many references to the original contributions as suggested. In particular, the following references are cited now and included/discussed in the paper: Allam and Tuck (1984a), Allam and Tuck (1984b), Bethan et al. (1996), Dethof et al. (1999), Knollenberg et al. (1993), Richard et al. (2006), Rosenlof et al. (1997), Tuck et al. (1997), Tuck et al. (2003), Tuck et al. (2004), and Vaughan and Timmis (1998).

*lines 35 et seq: This section misses out some earlier references that deal with research quality observations, and which say some relevant things about the tropical tropopause [1,2,3,4,5]. In particular, the average qw at the sea surface in the tropics is 355 K, significantly lower than the potential temperature of the tropical tropopause, however that is defined. Transport from the midlatitude stratosphere and local descent must contribute [3]. Reference [5] below has some highly relevant discussion, also applicable in large parts of the subsequent text. The original point about the role of the monsoon was made in [6], amplified in [7]. They should be referenced.*

We agree that earlier references should be referenced in the revised version. In particular, we have now cited a number of papers in the revised introduction (see above and references) as suggested.

*lines 45-50: It would be helpful to the reader if the location of the StratoClim campaign was supplied.*

We agree – the StratoClim campaign in 2017 was based in Kathmandu/Nepal

as it is now stated in the paper (see also reply to review 2).

*line 69: In 1972 the NASA WB57F flew around a Cb top and anvil penetrating the tropopause near Amarillo, Texas. It saw higher concentrations of water vapour downwind than upwind of the penetrating top and anvil. It is reported in the CIAP report, published by the US Dept of Transportation in 1974.*

We agree. There is a paper (Shlanta and Kuhn, 1973) describing the aircraft flight in question (around a thunderstorm top near Amarillo, Texas, in 1972). In this flight (and others) the local ozone concentration and the water vapour overburden were measured near thunderstorms that reached or penetrated the tropopause. Shlanta and Kuhn (1973) thus conclude that thunderstorms inject substantial amounts of water vapour into the stratosphere. The reference (Shlanta and Kuhn, 1973) is now cited and the flight is mentioned in the paper.

*lines 73-79: These are all model based arguments. Observational ones, considerably better founded, are in [5] and [8], in which the TTL (not then called that) was pointed out.*

We agree – the following text was added to the paper. "The existence of a TTL (but not using this term) was discussed earlier based on observations (Tuck et al., 1997; Richard et al., 2006)".

*line 118: Saturation over ice is a crude approximation, see discussion in [5].*

We agree. The reviewer is alluding to the differences between hexagonal and cubic ice (Murphy and Koop, 2005). Given the focus of this paper we have refrained from an extensive discussion of the issue, but we changed the text to: "... vapour in ice clouds is approximated as saturation over ice (see also the discussion in Richard et al., 2006)"; that is we refer now to reference [5].

*lines 133-140: Isotopic studies are very useful in principle. One process that needs discussion is the very fast exchange rates at aqueous particle surfaces. For example, an O atom in a water molecule that enters a particle may not be attached to the same H (and D) atoms when it leaves. Particles are not at equilibrium; Henry's Law does not apply.*

We agree with the importance of the discussed process. However, given the fact that the ChiWIS measurements are not directly used in this paper (see l. 136) we have refrained from a discussion of the issue. We have not changed the text here.

*lines 163-166: Such trajectories are Lagrangian sampling of an Eulerian system, so are not truly following a given unmixed air mass.*

We agree and have removed "ideally", it is stated in the paper now "It would be better, considering the evolution of an air parcel in a Lagrangian sampling, when interpreting measurements at a particular point in the lower stratosphere ...".

*lines 200-207: See the discussion in [9] as referenced in [5].*

We have added the missing information at those lines: "However, small and

large ice particles in the same air mass do not develop independently. It was suggested (Knollenberg et al., 1993; Richard et al., 2006) that the ablation of small ice particles by solar near-infrared radiation plays a role in the production of the lowest, unsaturated water vapour values; the vapour molecules from the ablated small particles distill over to the larger particles that have significant fall speeds (Müller and Peter, 1992)".

We also added the following text citing [5]: "Such observations were made earlier through measurements in flights between the surface and 18 km in late January 2004 from Costa Rica (10°N, Richard et al., 2006)".

*lines 272-281: Interesting. In general, for the whole paper, I find it surprising that there is no mention of the subtropical jet stream and its migration north during boreal summer. See [10].*

The subtropical jet stream is now mentioned in the introduction also in conjunction with the papers by Rosenlof et al. (1997) and Dethof et al. (1999). On the other hand, the focus of the paper is on the Geophysica measurements during StratoClim in 2017, which do not provide much information on the subtropical jet stream. Therefore there is no further extensive discussion here.

*lines 283-288: Note that an early stratosphere-troposphere GCM predicted the role of monsoonal circulation in air entry to the stratosphere [12].*

Thanks! This information is added to the introduction and section 4 of the paper, where it is stated: "It should be noted that an early stratosphere-troposphere model already predicted the role of monsoonal circulation in air entry to the stratosphere (Allam and Tuck, 1984a,b)".

*lines 318 & 332: "Control" is a slippery concept, given the nonconvergent variance of airborne observations [11].*

We agree that we should avoid the word "control"; it is stated now in the paper: "...water vapour mixing ratios in the Asian summer monsoon region are regulated by cold point temperatures ..." and that "...the location of the lapse rate tropopause in the Asian summer monsoon is thus *not* determined by local processes".

*lines 334-338: See Figure 14 in [3].*

Thanks for the comment. Figure 14 in Tuck et al. (2003) shows a scatterplot of the potential temperature at the tropopause, versus latitude for the WB57F flights in 1998 and 1999. Some flights showed a much lower midlatitude tropopause underlying the higher tropical tropopause at latitudes poleward of 23°N. – As the focus here is on the ASMA and information on the midlatitude tropopause poleward of 23°N was not provided by the Geophysica measurements during the campaign in question, we decided to not add this information to the conclusions of the paper.

**References**

Allam, R. J. and Tuck, A. F.: Transport of water vapour in a stratosphere-troposphere general circulation model II. Trajectories, Q. J. R. Meteorol. Soc., 110, 357–392, 1984a.

Allam, R. J. and Tuck, A. F.: Transport of water vapour in a stratosphere-troposphere general circulation model I. Fluxes, Q. J. R. Meteorol. Soc., 110, 321–356, 1984b.

Bethan, S., Vaughan, G., and Reid, S. J.: A comparison of ozone and thermal tropopause heights and the impact of tropopause definition on quantifying the ozone content of the troposphere, Q. J. R. Meteorol. Soc., 122, 929–944, 1996.

Dethof, A., O'Neill, A., Slingo, J. M., and Smit, H. G. J.: A mechanism for moistening the lower stratosphere involving the Asian summer monsoon, Q. J. R. Meteorol. Soc., 556, 1079–1106, 1999.

Knollenberg, R. G., Kelly, K. K., and Wilson, J. C.: Measurements of high number densities of ice crystals in the tops of tropical cumulonimbus, J. Geophys. Res., 98, 8639–8664, 1993.

Müller, R. and Peter, T.: The numerical modelling of the sedimentation of polar stratospheric cloud particles, Ber. Bunsenges. Phys. Chem., 96, 353–361, 1992.

Murphy, D. M. and Koop, T.: Review of the vapour pressures of ice and supercooled water for atmospheric applications, Q. J. R. Meteorol. Soc., 131, 1539–1565, 2005.

Richard, E. C., Tuck, A. F., Aikin, K. C., Kelly, K. K., Herman, R. L., Troy, R. F., Hovde, S. J., Rosenlof, K. H., Thompson, T. L., and Ray, E. A.: High-resolution airborne profiles of $CH_4$, $O_3$, and water vapor near tropical Central America in late January to early February 2004, J. Geophys. Res., 111, D13304, https://doi.org/10.1029/2005JD006513, 2006.

Rosenlof, K. H., Tuck, A. F., Kelly, K. K., Russell III, J. M., and McCormick, M. P.: Hemispheric asymmetries in the water vapor and inferences about transport in the lower stratosphere, J. Geophys. Res., 102, 13 213–13 234, https://doi.org/10.1029/97JD00873, 1997.

Shlanta, A. and Kuhn, P. M.: Ozone and water vapor injected into the stratosphere from two isolated thunderstorms, J. App. Meteorol. Climatol., 12, 1375–1378, URL https://doi.org/10.1175/1520-0450(1973)012<1375:OAWVII>2.0.CO;2, 1973.

Tuck, A. F., Baumgardner, D., Chan, K. R., Dye, J. E., Elkins, J. W., Hovde, S. J., Kelly, K. K., Loewenstein, M., Margitan, J. J., May, R. D., Podolske, J. R., Proffitt, M. H., Rosenlof, K. H., Smith, W. L., Webster, C. R., and Wilson, J. C.: The Brewer-Dobson circulation in the light of high altitude in situ aircraft observation, Q. J. R. Meteorol. Soc., 123, 1–69, 1997.

Tuck, A. F., Hovde, S. J., Kelly, K. K., Mahoney, M. J., Proffitt, M. H., Richard, E. C., and Thompson, T. L.: Exchange between the upper tropical troposphere and the lower stratosphere studied with aircraft observations, J. Geophys. Res., 108, 4734, https://doi.org/10.1029/2003JD003399, 2003.

Tuck, A. F., Hovde, S. J., Kelly, K. K., Reid, S. J., Richard, E. C., Atlas, E. L., Donnelly, S. G., Stroud, V. R., Cziczo, D. J., Murphy, D. M., Thomson, D. S., Elkins, J. W., Moore, F. L., Ray, E. A., Mahoney, M. J., and Friedl, R. R.: Horizontal variability 1–2 km below the tropical tropopause, J. Geophys. Res., 109, https://doi.org/10.1029/2003JD003942, 2004.

Vaughan, G. and Timmis, C.: Transport of near-tropopause air into the lower midlatitude stratosphere, Q. J. R. Meteorol. Soc., 124, 1559–1578, 1998.